# Biophysical Characterization of Membrane Proteins Embedded in Nanodiscs Using Fluorescence Correlation Spectroscopy

**DOI:** 10.3390/membranes12040392

**Published:** 2022-03-31

**Authors:** Matthew J. Laurence, Timothy S. Carpenter, Ted A. Laurence, Matthew A. Coleman, Megan Shelby, Chao Liu

**Affiliations:** 1Biosciences and Biotechnology Division, Lawrence Livermore National Laboratory, Livermore, CA 94550, USA; laurence3@llnl.gov (M.J.L.); carpenter36@llnl.gov (T.S.C.); coleman16@llnl.gov (M.A.C.); 2Materials Science Division, Lawrence Livermore National Laboratory, Livermore, CA 94550, USA; laurence2@llnl.gov; 3Department of Radiation Oncology, University of California Davis, Sacramento, CA 95616, USA

**Keywords:** fluorescent correlation spectroscopy, membrane proteins, nanodiscs, cell-free expression

## Abstract

Proteins embedded in biological membranes perform essential functions in all organisms, serving as receptors, transporters, channels, cell adhesion molecules, and other supporting cellular roles. These membrane proteins comprise ~30% of all human proteins and are the targets of ~60% of FDA-approved drugs, yet their extensive characterization using established biochemical and biophysical methods has continued to be elusive due to challenges associated with the purification of these insoluble proteins. In response, the development of nanodisc techniques, such as nanolipoprotein particles (NLPs) and styrene maleic acid polymers (SMALPs), allowed membrane proteins to be expressed and isolated in solution as part of lipid bilayer rafts with defined, consistent nanometer sizes and compositions, thus enabling solution-based measurements. Fluorescence correlation spectroscopy (FCS) is a relatively simple yet powerful optical microscopy-based technique that yields quantitative biophysical information, such as diffusion kinetics and concentrations, about individual or interacting species in solution. Here, we first summarize current nanodisc techniques and FCS fundamentals. We then provide a focused review of studies that employed FCS in combination with nanodisc technology to investigate a handful of membrane proteins, including bacteriorhodopsin, bacterial division protein ZipA, bacterial membrane insertases SecYEG and YidC, *Yersinia pestis* type III secretion protein YopB, yeast cell wall stress sensor Wsc1, epidermal growth factor receptor (EGFR), ABC transporters, and several G protein-coupled receptors (GPCRs).

## 1. Isolation of Membrane Proteins

Membrane protein research is critical to understanding cell uptake (e.g., of nutrients, pharmaceuticals, invading pathogens and toxic materials), signaling across the membrane, and diverse other functions. Unfortunately, the hydrophobicity of membrane proteins contributes to their insolubility and instability outside their natural lipid bilayer environment, thus complicating their purification and subsequent structural and solution-based studies. Indeed, while they comprise almost 30% of all human proteins [1], membrane proteins make up only 3–4% of structures in the Protein Data Bank [2].

While the common approach of detergent micelle solubilization can enable solution studies of membrane proteins, these methods are fraught with difficulties in producing suitable quantities of pure, stable, soluble, monodisperse protein and often require truncation or modification of the full-length membrane protein to create a stable construct. Optimization of solubilization conditions while maintaining native fold can be difficult to impossible and may lead to the loss of functionally critical phospholipids, making aggregation challenging to avoid [3]. Newer amphiphilic reagents with advantages over conventional detergents continue to be developed, including the use of peptides [4,5,6,7], amphiphilic polymers [8], and maltose-neopentyl glycol (MNG) amphiphiles [9]. Alternatively, modifying the membrane protein of interest can also enable solution-based studies, for example by fusion with an N-terminal peptide that prevents membrane insertion and with a C-terminal amphipathic protein that keeps it soluble [10]. However, all of the above approaches remove the membrane protein of interest from its native lipid environment, which can significantly impact protein conformation and function.

As a result, a major focus of research has been the development of membrane mimetics that can be used to assess the in vitro behavior of membrane proteins while preserving their native structure and function, the most widespread of which are liposomes [11,12]. Unlike detergents, liposomes provide a more native environment in which membrane proteins are embedded in a lipid bilayer. They can be made in different sizes ranging from 20 nm to >1 µm depending on the preparation method [13,14,15,16]. However, liposomes present their own drawbacks and suffer from substantial issues with size heterogeneity and reproducible production of high-quality liposomes for different protein systems [13,17]. Liposomes are also often sparingly soluble and provide no control over how individual proteins inserted interact and oligomerize, thus confounding studies of protein association and binding that require isolated entities.

## 2. Nanodiscs as a Membrane Mimetic

An alternative method to simulate the membrane environment in solution is the use of membrane mimetics that form membrane protein supporting scaffolds in the form of a lipid “nanodisc”. Nanodiscs consist of a section of lipid bilayer ~10 nm in diameter surrounded by a supporting belt or scaffold (Figure 1) [18]. Integral membrane proteins can be embedded in the bilayer to form soluble protein–nanodisc complexes. The composition of the bilayer can be varied to contain specific lipid mixtures, cholesterol, lipopolysaccharides, and synthetic lipids or membrane-incorporating species such as telodendrimers [19,20]. Thus, the composition can be specified to mimic cellular membranes, while species with specific functionality (e.g., biotinylation [21,22,23,24], Ni functionalization [25,26,27,28] or fluorescent labels) can also be incorporated. In some cases, lipids from native membranes can also be incorporated [29,30,31,32,33,34,35]. Compared to lipid vesicles, nanodiscs are easier to synthesize with monodisperse or discrete size distributions and do not introduce any unnatural curvature. Multiple types of nanodiscs have been developed in recent decades which principally differ based on the scaffold used to contain the lipid bilayer, including scaffolds consisting of apolipoproteins (to form nanolipoprotein particles, or NLPs), polymers, saposin, peptides, and DNA. Each platform has advantages and drawbacks for use in structural and biochemical studies, as discussed in a recent review [36].

Nanolipoprotein particles (NLPs), the most widely adopted version of nanodiscs, use membrane scaffold proteins (MSPs) to encircle and stabilize the lipid bilayer. The MSP usually consists of two copies of an apolipoprotein, an amphipathic α-helical protein with a physiological role in lipid-binding and transport. NLPs were first developed in 2002 by Bayburt, Grinkova, and Sligar while studying human membrane-binding proteins as a means to create realistic and structurally consistent artificial membranes [37]. Common MSPs use a truncated version of the human lipid-binding protein Apo A-1 (MSP1) and its various engineered derivatives, producing nanodiscs with diameters ranging from 10 to 15 nm [3,19,38]. More recently, Nasr et al. achieved even greater homogeneity in nanodisc size (11–12 nm) by covalently linking the C- and N-termini of MSPs. These “covalently circularized nanodiscs” have improved thermostability and proteolytic resistance, and the technology can be extended to produce a variety of defined sizes and geometries [39,40].

However, a limitation of MSP-based nanodiscs is that both membrane protein and apolipoprotein must be recombinantly expressed and solubilized by detergents before they are assembled into nanodiscs, which may cause denaturation and aggregation before the membrane protein of interest can be introduced into the membrane mimetic. Cell-free co-translation of the scaffold protein and target membrane protein in the presence of lipid, which leads to the embedding of the membrane protein into the bilayer and self-assembly of apolipoprotein scaffolded nanodiscs, enables the encapsulation of membrane proteins within homogenous populations of membrane mimetics particles while circumventing the need for detergent solubilization and purification before lipid bilayer insertion [41,42,43]. This avoids often lengthy optimization steps, both in detergent selection and in the engineering of stable constructs, to ensure the membrane protein is soluble and presents a native fold. Importantly, this both functionally expands the utility of nanodisc encapsulation to full-length membrane proteins previously not amenable to detergent solubilization and preserves native function for such systems [44,45].

Another limitation of NLPs is that depending on the method of preparation, some heterogeneity in disc size and membrane protein insertion rate can remain. One study noted a broad size distribution [46], while several others reported the existence of the inserted membrane protein in a multimeric state [29,47]. Greater control over size and protein insertion rate can be achieved by various techniques and should be optimized for the particular protein under study. For example, with the ability to precisely prescribe the size of covalently circularized nanodiscs and lipid compositions, one can influence the number of proteins incorporated in the constrained bilayer [40]. It has also been demonstrated that with careful specification of plasmid levels and lipid compositions in cell-free expression systems, a monomeric incorporation of the membrane protein can be achieved in ~80% of NLPs [48]. Such techniques can be tuned to reduce heterogeneity and specify insertion rate as suitable for different types of studies.

Other varieties of nanodisc have been developed that use a co-polymer rather than lipid-binding proteins as the surrounding belt, as reviewed recently [34,49,50]. The most widely used polymer is styrene maleic acid (SMA), first shown to form nanodiscs on introduction to bacteriorhodopsin loaded liposomes by Knowles et al. in 2009 [30]. Newer co-polymers such as di-isobutylene maleic acid (DIBMA) are increasingly utilized to vary disc geometry and tune polymer/membrane interaction [51,52]. SMA removes a section of lipid bilayer to form styrene co-maleic acid lipid particles (SMALPs), a technique that has found greatest use in detergent-free solubilization of proteins of interest from native cell membranes for biochemical and structural studies [31,32,33,34,35,53,54,55]. Avoiding disassembly and reconstitution allows membrane proteins to exist in a disc of membrane as close as possible to their natural cellular conditions. SMALPs also display excellent thermal and temporal stability, keeping their protein monodisperse and functional for over a week at 4 °C [35,56]. While polymer nanodiscs do not have the benefit of the discretized size population of MSP-based nanodiscs, varying the co-polymer monomer ratios and new polymer designs are improving their size control, pH stability, and other spectroscopic and biochemical properties [57].

## 3. Fluorescence Correlation Spectroscopy

The interactions between membrane proteins and their ligands, agonists, and antagonists provide a basis for understanding their physiological functions and how they are affected by pathogens, cancer, and drugs. Techniques for studying these interactions include atomic force microscopy (AFM), surface plasmon resonance (SPR), circular dichroism spectroscopy, and fluorescence techniques such as fluorescence correlation spectroscopy (FCS), fluorescence recovery after photobleaching (FRAP), and Förster resonance energy transfer (FRET). Here, we first focus on FCS, then review the use of FCS combined with nanodisc technologies in studies of several membrane proteins.

In comparison to other techniques, the relative strength of fluorescence techniques are their specificity, sensitivity, and high-throughput potential. While the species of interest must be labeled with chromophores or bulky fluorescent proteins, this ensures that the signal observed is from the molecule of interest. In addition, multiple colors may be used, allowing multiple species to be traced simultaneously in the same sample. Furthermore, fluorescence techniques can be used in tandem with other forms of microspectroscopy [58].

Fluorescence correlation spectroscopy is one of the most common fluorescence techniques to study molecular dynamics (Figure 2). In its most common configuration, a confocal microscope is used to create a femtoliter-sized detection volume in a solution of fluorescent tagged molecules [59,60]. Individual photons emitted from the fluorescent molecules diffusing through the detection volume are detected using single photon counting detectors, which record the timestamps of the photon arrivals [61]. By performing the mathematical correlation operation, the data are converted into a correlation curve which can be fit to models for various diffusion processes.

FCS presents several advantages over related fluorescence systems. Fluorescence recovery after photobleaching (FRAP) measures protein diffusion kinetics by briefly photobleaching a region inside a solution, cell, or membrane containing fluorescently labeled proteins and measuring the time needed for the region to recover the fluorescence signal as non-bleached proteins diffuse in. Unlike FRAP, FCS can measure multicomponent diffusion and protein concentrations, requires lower protein concentrations, and is better suited for short time scale observations [62,63]. Förster resonance energy transfer (FRET) is a form of nonradiative energy transfer between two molecules used to determine if they are in proximity to each other. As the FRET efficiency is proportional to the sixth power of distance between the fluorescent donor and acceptor, FRET is a very sensitive method to probe for intermolecular binding and protein conformations [64,65,66,67]. In practice, however, binding measurements using FRET are much more challenging than using FCS because successful design of FRET pairs at specific positions on proteins requires precise knowledge of protein structures.

By attaching fluorescent tags to biological macromolecules, virtually any molecule in solution can be studied by FCS. Proteins can be either expressed as fusions with other fluorescent proteins or functionalized with chemical fluorescent dyes. The sample does not need to be removed from solution, allowing solution dynamics to remain unchanged. In addition to measurements in solution, FCS may be measured inside cells [68,69,70,71] allowing findings from simpler solution-based assays to be validated in cell-based experiments.

Analysis of the correlation curve from FCS measurements gives insight into the physical properties of a system. The shape of the correlation curve depends on an effective detection profile produced by the laser excitation volume combined with the detection pinhole. The simplest model for FCS is that of a single species diffusing in a 2D Gaussian detection profile, with correlation *C* given by
(1)$$C\left(\tau \right)=1+\frac{1}{N\left(1+\frac{\tau }{\tau_{D}}\right)}$$
where τ is the time delay,$\;\tau_{D}$ is the diffusion time, and *N* is the average number of fluorescent molecules in the detection volume. $\tau_{D}$ represents the average time for a molecule to diffuse out of the open detection volume in the solution. The diffusion constant *D*, which is dependent on factors including molecular size and structure, can be calculated by $D=\frac{r^{2}}{4\tau_{D}}$, where r is the width of the 2D Gaussian profile as determined through a calibration. The concentration of the species can be calculated from *N* and the calibrated volume. In this model, diffusion in the axial dimension (along the optical path) is ignored since the timescale of diffusion out of the open detection volume in that axis is much longer.

Deviations from the simple model above can indicate the presence of additional species with different diffusion times [72], an anomalously diffusing species, or other photophysical behaviors such as triplet-state excitation [73]. The diffusion of two distinct species with average numbers *N*_1_ and *N*_2_ and diffusion times $\tau_{D1}$ and $\tau_{D2}$ can be modeled with correlation function
(2)$$C\left(\tau \right)=1+\frac{N_{1}}{{\left(N_{1}+N_{2}\right)}^{2}\left(1+\frac{\tau }{\tau_{D1}}\right)}+\frac{N_{2}}{{\left(N_{1}+N_{2}\right)}^{2}\left(1+\frac{\tau }{\tau_{D2}}\right)}$$

The presence of multiple species may be due to different entities (e.g., different proteins) or different states of the same entity (e.g., oligomers or aggregates). The presence of two species is expected when a biomolecule is present that binds to the fluorescently tagged species, for example in cases of two interacting proteins or drug binding to protein. Under sub-saturating concentrations of the non-fluorescent molecule, one species would be the bound complex (with a longer diffusion time) while the second would be the free fluorescently labeled molecule. A shift in the correlation curve towards longer diffusion times is a simple readout of binding when probing for molecular interactions. Note, however, that the diffusion times of the two species must differ by at least a factor of 1.6 to be distinguishable under typical conditions [72]. Since diffusion time is inversely proportional to the cubed root of mass, this criterion stipulates that the masses of the two species must differ minimally by a factor of 4. Thus, a critical strategy when investigating potential interactions (e.g., during drug screening) is the placement of the fluorescent label on the small ligand so that a large change in mass occurs upon binding of the larger protein.

FCS can also be expanded to use multiple colors to observe multiple molecular species simultaneously. In fluorescence cross-correlation spectroscopy (FCCS), two or more molecular species that may interact are labeled with spectrally dissimilar fluorescent tags and excited with separate lasers [74]. The distinct fluorescence signals can be autocorrelated individually as in single-color FCS or cross correlated. Fitting the cross-correlation curve gives the concentration of bound pairs, which can be used with the concentrations of each unbound species to calculate the binding coefficient [61]. Alternating laser excitation and pulsed-interleaved excitation allow for the elimination of many sources of spurious cross-correlations [75,76,77].

Conventional FCS only measures diffusion at a single spot of hundreds of nanometers to 1 µm in size, much smaller than many target areas of interest in biological systems, thus missing potential spatial heterogeneity in the sample. Various methods have been developed that extend conventional FCS to provide greater spatial information. Scanning FCS performs measurements as the observation volume is continuously moved through the sample, providing spatial information while reducing photobleaching [78]. Imaging correlation spectroscopy makes use of EMCCD cameras that achieve single-photon sensitivity over a much wider area. Instead of tracking photon counts from a single point, the camera records an image containing FCS data for hundreds or thousands of pixels simultaneously [79,80]. Acquiring many correlations in parallel across space not only provides location-dependent information but also increases the statistics while reducing acquisition time, thus allowing high-throughput measurements of diffusing species and their interactions.

FCS has several limitations as a technique in general and when applied to the study of membrane proteins. As with all fluorescence techniques, one must first ensure that the fluorescent label does not interfere with the structure or function of the protein under study. The requirement that two species have masses that differ by at least a factor of 4 to be distinguishable in single-color FCS has been discussed above [72]. Accurate concentration determination requires an accurate calibration of the detection volume as well as sufficient signal-to-noise ratio. While the use of higher laser powers improves the signal-to-noise ratio, it can also lead to triplet-state excitation and enlargement of the detection volume [73,81,82]. High laser powers can also lead to photobleaching of fluorophores, a common issue with many fluorescence measurements. Photobleaching is especially detrimental in experiments involving slowly diffusing molecules that require long acquisition times, in which case lower laser powers should be used. In addition, slowly diffusing aggregations of fluorescently labeled proteins, a problem common with membrane proteins, produce strong signals that can dramatically skew correlation curves even with a comparatively low concentration of aggregates. This again underscores the importance of methods such as nanodiscs to producing soluble non-aggregated membrane proteins. Finally, proper interpretation of FCS data requires fitting to a suitable model. Photobleaching, triplet-state excitation, aggregations, and complexities arising from measurements taken inside cells or on membranes (e.g., constrained diffusion, immobilization, and active transport) all result in correlation curves requiring models beyond simple diffusion [83,84].

## 4. FCS Measurements of Membrane Proteins Embedded in Nanodiscs

Nanodiscs permit the application of solution-based techniques such as FCS to the study of individual membrane proteins embedded in well-defined bilayers. Starting in the past decade, FCS has been applied to the study of various membrane proteins embedded in nanodiscs, as reviewed below and summarized in Table 1.

In the first FCS characterization of a nanodisc-embedded membrane protein, Gao et al. measured the diffusion times of bacteriorhodopsin (bR) embedded in nanodiscs made of a DMPC-lipid bilayer surrounded by truncated Apo-A1 [47]. Empty and bR-loaded nanodiscs had diffusion times ~0.5 ms. The authors also measured the diffusion times of several particles (e.g., large vesicles, beads, and fluorescent dyes) of known sizes as determined by dynamic light scattering. From this calibration of diffusion time vs. particle size, the ~0.5 ms diffusion times of empty and bR-nanodiscs correspond to diameters of ~10 nm, agreeing with the value expected of Apo-A1 nanodiscs.

Since then, FCS has been used to characterize interactions between several other nanodisc-embedded bacterial membrane proteins and their soluble binding partners. Hernandez-Rocamora et al. measured the binding of nanodisc-embedded *E. coli* ZipA, which provides essential membrane tethering during cell division, to FtsZ in different polymerization and nucleotide states [85]. Empty nanodiscs made of *E. coli* polar lipid extract surrounded by MSP1D1 and discs containing one ZipA protein had diffusion times of ~0.5 ms, corresponding to diameters of 10 and 13 nm, respectively, as expected. The addition of FtsZ resulted in a rightward shift of the correlation curve from which the fraction of bound species was determined.

Wu et al. [86], Kedrov et al. [87], and Geng et al. [88] elucidated the specific nature of binding between the ribosome and *E. coli*’s two membrane insertases, SecYEG and YidC and, in the process, revealed the importance of using a native-like lipid environment when attempting to uncover physiological binding behavior. Membrane insertases associate with translating ribosomes and integrate nascent polypeptides into the lipid bilayer. By FCS measurements of SecYEG embedded in nanodiscs, Wu et al. found that SecYEG binds ribosome strongly only in the presence of a nascent chain (ribosome-bound nascent chain complex, RNC) as seen by a rightward shift in the diffusion curve [86]. This requirement of nascent chain was not fully apparent if not for the use of nanodiscs, as ribosome without nascent chain was still able to bind the majority of SecYEG solubilized in detergent. The authors further found that the addition of the SecA protein, which also binds SecYEG, competitively inhibits binding of RNC to SecYEG-nanodiscs, as seen by decreases in the fraction of RNC-SecYEG-nanodisc species in FCS diffusion curves. In contrast, this competitive binding behavior of SecA and RNC to SecYEG was not observed when SecYEG was solubilized in detergent rather than nanodiscs; in detergent, almost all SecYEG was bound to RNC despite the presence of excess SecA. In a parallel study to that of SecYEG, Kedrov et al. found that when embedded in nanodiscs, YidC binding to ribosome, as measured by a rightward shift in the FCS correlation curve, also requires the presence of a translating nascent chain [87]. In contrast, YidC in detergent exhibited several artifactual interaction characteristics, such as binding of ribosome without nascent chain at non-physiological acidic pH. Geng et al. further explored the structural basis of the binding between YidC-nanodiscs and RNC through FCS measurements of YidC mutant constructs and found that YidC’s cytosolic loop C2 and C terminus are both crucial in this interaction [88]. All the above results underscore the importance of nanodiscs in providing a native-like environment to study physiological membrane protein interactions.

Ly et al. used FCS to measure the interactions between *Yersinia pestis* membrane-bound YopB and soluble LcrV, two proteins of the bacteria’s type III secretion system critical for host cell invasion [89]. Binding of YopB-nanodisc to eGFP-LcrV resulted in a rightward shift in the correlation curve from which the fraction of bound species was determined (Figure 3). A full binding curve obtained by varying concentrations of YopB-nanodisc produced a dissociation constant ~20 nM. In addition to enabling FCS measurements, YopB produced concurrently with nanodiscs in the cell-free expression system had the added advantage of greater protein solubility [44].

In addition to studying the above bacterial proteins, FCS in combination with nanodisc technology has been used to characterize various eukaryotic membrane proteins. Voskoboynikova et al. used FCS to demonstrate successful production of monodisperse Wsc1, a yeast transmembrane cell wall stress sensor, embedded in SMALPs by SMA extraction from native *S. cerevisiae* cell membrane [92]. Other studies further characterized by FCS the interactions between eukaryotic membrane proteins and their substrates. Quinn et al. measured FCS on nanodisc-embedded full-length epidermal growth factor receptor (EGFR) produced by cell-free expression [48]. Remarkably, the authors reported the ability to detect microsecond shifts in diffusion times upon ATP binding to EGFR-nanodisc and upon phosphorylated EGFR-nanodisc binding to a specific antibody. Li et al. [90] and Horsey et al. [29] used FCS to study ligand binding of ATP binding cassette (ABC) transporters, which couple ATP hydrolysis to the binding and transport of substrates across membranes and which play major roles in drug interactions and resistance. Li et al. found that the binding of several fluorescent-labeled substrates to nanodisc-embedded ABCB1/P-glycoprotein is dependent on different nucleotide states of the protein [90]. Horsey et al. measured the binding of fluorescent-prazosin to ABCG2/BCRP (breast cancer resistance protein) embedded in SMALP by SMA extraction from the native HEK293 cell membrane [29].

Finally, FCS combined with nanodisc technology has been used to study G protein-coupled receptors (GPCRs), the largest family of membrane proteins in humans and major drug targets. Dathe et al. purified SMALPs containing neurotensin receptor 1 (NTSR1) and found its diffusion time by FCS to be much larger than expected along with large fitting residuals, suggesting a broad rather than monodisperse distribution of sizes [46]. Other studies investigated ligand binding to GPCRs. Gao et al. produced several nanodisc-bound human GPCRs by cell-free expression, including neurokinin-1 receptor (NK1R), adrenergic receptor ADRB2, and dopamine receptor DRD1 [45]. The authors proceeded to measure by FCS a full binding curve between NK1R-nanodisc and its fluorescently labeled ligand Substance P, from which a dissociation constant ~83 nM was obtained. More recently, Grime et al. measured the binding of human adenosine A_2A_ receptor embedded in SMALP to its fluorescently labeled antagonist xanthine amine congener [91]. The use of FCS to measure interactions between a small fluorescently-labeled ligand and a large nanodisc-embedded GPCR invites many future studies for potential drug development.

Beyond the determination of the binding and diffusion biophysics of membrane proteins, the combination of FCS and nanodisc technology has been used to study the protonation effect of membranes. Lipid membrane can enrich [H^+^] near its surface in a phenomenon known as the local membrane proton-collecting antenna effect, serving to accelerate proton uptake by membrane-bound proton transporters such as bacteriorhodopsin and cytochrome *c* oxidase (Cyt*c*O). Xu et al. measured FCS curves of fluorescein-labeled nanodiscs of different sizes, in the absence or the presence of membrane-embedded Cyt*c*O, and under various buffer conditions [93]. The curves were fitted to a multistate model with components including normal diffusion and fluorescence decay, the latter serving as the readout of protonation levels as fluorescein fluorescence is highly sensitive to pH. While previous studies had measured the proton antenna effect for large 30 nm vesicles, the use of nanodiscs allowed the authors to detect the effect for membranes as small as 9 nm in diameter. The authors also found that the presence of Cyt*c*O in the membrane decreased the effect, requiring a larger membrane (12 nm in diameter) to achieve the same protonation levels.

A direct comparison of diffusion times is not possible across all the above studies since different FCS setups have slightly different focal volumes. Rather, conclusions are drawn within a study based on changes in correlation curves under different biological conditions. Nevertheless, it is informative to visualize FCS curves of nanodiscs-membrane proteins in relation to other species across a wide range of sizes. As expected, correlation curves and diffusion times of membrane proteins embedded in ~10 nm-nanodiscs lie between those of individual fluorescent proteins (GFP and mCherry) and polystyrene microspheres with diameters ~40–100 nm (Figure 4).

## 5. Summary and Future Directions

Despite its age, FCS remains an effective and widely used biophysical technique to study protein biophysics and binding in solution as well as within cells. Nanodiscs enable FCS measurements of in-solution, purified, individual membrane proteins residing in a native-like lipid environment. While only a handful of studies have used this combination as reviewed above, the simplicity of FCS combined with the versatility of nanodiscs presents great potential for future studies of the plethora of membrane proteins. FCS may be applied to any of the ~100 membrane proteins that have been successfully reconstituted in nanodiscs (as listed in several recent reviews [3,34]) as well as important nanodisc-embedded membrane proteins to be successfully isolated in the future. Candidates for the latter include other bacterial type III secretion proteins which play critical roles in host cell invasion, in FCS binding studies analogous to those performed for *Y. pestis* YopB [89].

Future studies may leverage the ability to precisely control lipid composition in nanodiscs to investigate the effects of lipids on membrane protein function and structure. Membrane lipids have immense diversity, with the human body containing thousands of different lipids with significant implications for cellular shape and function [94,95]. Studies have broadly established that membrane lipid composition, such as cholesterol content and phospholipid type, affects membrane protein behavior through both direct lipid-protein interactions and indirect mechanisms such as membrane fluidity and curvature [96,97,98,99,100,101,102]. In recent years, computational research in lipid bilayers has also had a focus on increasing the complexity of the system to more closely replicate various biological membranes [103,104,105,106,107]. This coordinated drive towards complex realism in experimental and computational work opens the opportunities for both disciplines to inform the other. Of particular interest are studies of the major drug targets GPCRs embedded in nanodiscs of various prescribed lipid compositions. The effects of the surrounding membrane composition on interactions with G proteins, affinities for ligands, and affinities for therapeutic drugs of many GPCRs remain to be investigated by computational tools, FCS, and other experimental techniques.

The FCS-nanodisc assay shows great promise for future improvements. Advancements in detector technology will continue to increase the range of concentrations over which FCS can be performed by increasing photon count limits and time resolution [108,109]. New types of spatial extent FCS such as line scanning FCS or imaging FCS have not yet been applied to nanodisc measurements, but future work may find them useful in enhancing the throughput power of the assay over a short time interval. Continued improvements to nanodisc technology, such as further development and use of polymers and membrane scaffold proteins, will facilitate the ease of protein isolation and increase the fidelity of the mimetic to the native membrane. Future studies employing such nanodiscs will give FCS access to a broader selection of context-relevant membrane proteins and the complexes they form.

## Figures and Tables

**Figure 1 membranes-12-00392-f001:**
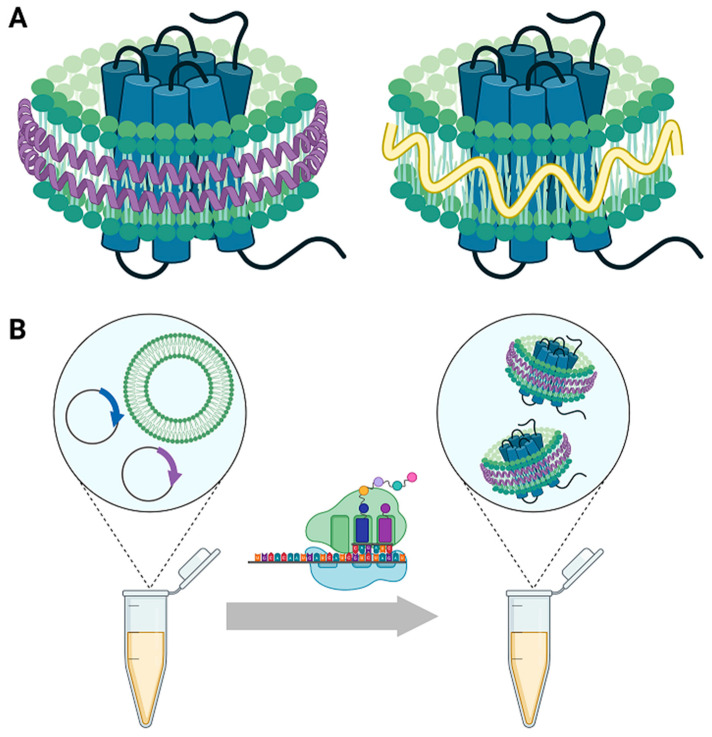
(**A**) Transmembrane protein embedded in a nanodisc lipid bilayer supported by (**left**) scaffold lipoproteins (NLPs) and (**right**) styrene maleic acid copolymers (SMALPs). (**B**) One-pot cell-free synthesis of membrane protein NLP complexes using a co-translational approach: transmembrane proteins and apolipoproteins are translated from cDNA simultaneously in the presence of liposomes using the translational machinery of an *E. coli* cell-free lysate, leading to the self-assembly of nanodisc loaded with the membrane protein of interest [41,42,43].

**Figure 2 membranes-12-00392-f002:**
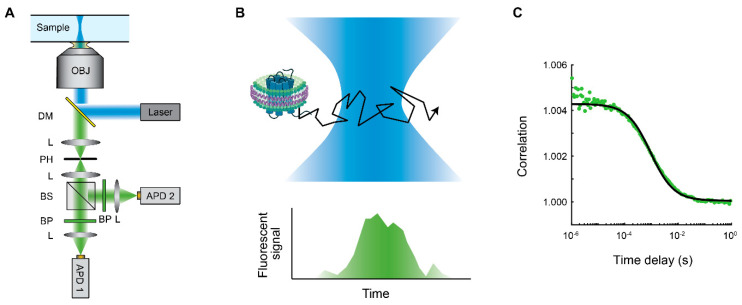
Fluorescence correlation spectroscopy (FCS) applied to membrane proteins embedded in nanodiscs. (**A**) FCS instrument schematic for a single-color system based on a confocal microscope. A tightly focused laser excitation combined with a confocal pinhole define a small, femtoliter-scale open detection volume. The emitted fluorescence signal is typically split and projected onto two detectors to eliminate detector dead time and afterpulsing effects. OBJ: objective; DM: dichroic mirror; L: lens; PH: pinhole; BS: beamsplitter; BP: bandpass filter; APD: avalanche photodiode. (**B**) As a molecule of the fluorescently labeled species, such as the membrane protein embedded in a nanodisc shown, diffuses through the open detection volume, the detected fluorescence signal fluctuates. Size of nanodisc in comparison to laser focus not drawn to scale. (**C**) The *g*^(2)^ correlation is calculated from the fluorescence signal (green data points). A simple diffusion model (Equation (1)) (black line) is fitted to the data, giving the diffusion time and concentration of the diffusing species. Data shown are of the human β-2 adrenergic receptor conjugated with the green fluorescent protein and embedded inside an Apo-A1 encircled DMPC nanodisc. Fitted diffusion time *τ*_D_ = 0.92 ± 0.02 ms.

**Figure 3 membranes-12-00392-f003:**
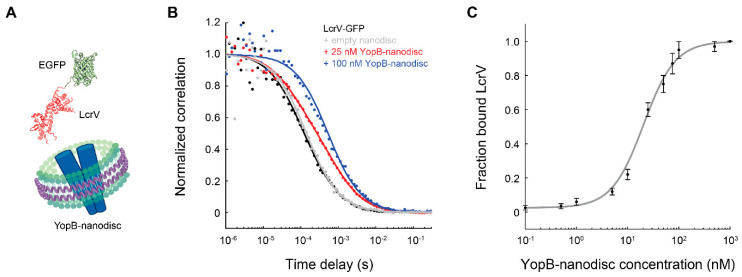
Binding of nanodisc-embedded membrane proteins to smaller, fluorescently labeled ligands are easily detectable by FCS due to a large change in diffusion time. (**A**) Cartoon of the *Y. pestis* membrane protein YopB (a multimer) inserted into a 10 nm nanodisc and its cognate protein LcrV (PDB: 1R6F) labeled with EGFP. (**B**) FCS correlation curves of LcrV-EGFP in the presence of different concentrations of YopB-nanodisc (data selected from Ly et al. [89]). Addition of YopB-nanodisc shifts the correlation curve of LcrV-EGFP to the right. Intermediate concentrations of YopB-nanodisc result in a diffusion curve with two species present, one of free LcrV-EGFP and one bound to YopB-nanodisc. Curves are fitted to Equation (2) to obtain the diffusion times and fractions of the two species. (**C**) Fraction of LcrV-EGFP bound to YopB-nanodisc at different YopB-nanodisc concentrations. A fit to the Hill equation gives dissociation constant *K*_D_ = 20 ± 2 nM and Hill coefficient *n* = 1.4 [89].

**Figure 4 membranes-12-00392-f004:**
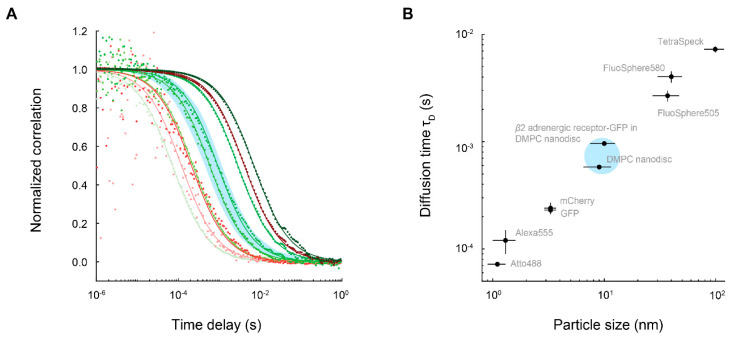
FCS correlation curves and diffusion times of nanodisc-embedded membrane proteins among a large range of particle sizes. (**A**) A representative correlation curve, from data collected for 1 min, for each species given in (**B**). Blue shaded area represents the range in which curves for membrane proteins embedded in nanodiscs are expected to reside. Green shades represent data collected with 488 nm laser excitation, and red shades represent data collected with 561 nm laser excitation. Data were collected on a homemade FCS system with a calibrated detection volume of ~3 µm^3^. (**B**) Diffusion times of different species plotted against particle size (diameter). Error bars on diffusion times are standard deviations based on multiple measurements. Error bars on size are estimated from diameters of the species. Blue shaded area represents the range in which nanodisc-embedded membrane proteins are expected to reside. Atto488 (Sigma-Aldrich, St. Louis, MS, USA, product #41051); Alexa555: Alexa Fluor 555 NHS ester (Thermo Fisher Scientific, Carlsbad, CA, USA, product #A20009); FluoSphere505 and FluoSphere580: FluoSpheres fluorescent carboxylate-modified microspheres 0.04 µm (Thermo Fisher Scientific, product #F10720); TetraSpeck: TetraSpeck fluorescent microspheres 0.1 µm (Thermo Fisher Scientific, product #T7279).

**Table 1 membranes-12-00392-t001:** Membrane proteins embedded in nanodiscs and studied by FCS.

Membrane Protein	Species	Expression and Assembly System	Nanodisc Type	Fluorescent Label	Diffusion Characteristic (FCS)	Reference
Bacteriorhodopsin (bR)	*H. salinarum*	Cell-free co-translation	DMPC + ApoΔ49A1	BODIPY-lysine during translation	*τ*_D_ = 0.35 ms	Gao et al., 2011 [47]
ZipA	*E. coli*	Proteins from *E. coli*, then mixed with lipids	*E. coli* lipid extract + MSP1D1	Lissamine rhodamine B on lipids	*D* = 32 ± 4 µm^2^/s	Hernandez-Rocamora et al., 2012 [85]
SecYEG	*E. coli*	Proteins from *E. coli*, then mixed with lipids	DOPC, DOPE, DOPG, CL + MSP1D1	Alexa Fluor 488 on protein	*D* = 27 ± 3 µm^2^/s	Wu et al., 2012 [86]
YidC	*E. coli*	Proteins from *E. coli*, then mixed with lipids	DOPC, DOPE, DOPG, CL + MSP1D1	Alexa Fluor 488 or Atto 647N on protein	*D* = 31 ± 2 µm^2^/s	Kedrov et al., 2013 [87]
YidC	*E. coli*	Proteins from *E. coli*, then mixed with lipids	DOPC, DOPE, DOPG + MSP1D1	Alexa Fluor 488 on protein	*D* = 39 ± 2 µm^2^/s	Geng et al., 2015 [88]
YopB	*Y. pestis*	Cell-free co-translation	DMPC + ApoΔ49A1	Binding to LcrV-GFP	*τ*_D_ = 0.63 ± 0.06 ms	Ly et al., 2014 [89]
Epidermal growth factor receptor (EGFR)	*H. sapiens*	Cell-free co-translation	DMPC + ApoΔ49A1	SNAP fusion construct labeled with SNAP surface 594	*τ*_D_ = 0.167 ± 0.002 ms	Quinn et al., 2019 [48]
ABCB1/P-glycoprotein	*M. musculus*	Membrane protein from *P. pastoris*, MSP from *E. coli*, then mixed with lipids	DMPC + MSP1D1	Binding to small BODIPY-linked ligands	*τ*_D_ = 3.0 ± 0.2 ms	Li et al., 2017 [90]
ABCG2/BCRP (breast cancer resistance protein)	*H. sapiens*	HEK293T expression, then native membrane extraction by SMA	SMALP	GFP fusion construct	*D* = 31 ± 4 µm^2^/s (likely dimer)	Horsey et al., 2020 [29]
Neurokinin-1 receptor (NK1R)	*H. sapiens*	Cell-free co-translation	DMPC + ApoΔ49A1	GFP fusion construct	*τ*_D_ = 0.51 ± 0.37 ms	Gao et al., 2012 [45]
Adenosine A_2A_ receptor (A2AR)	*H. sapiens*	*P. pastoris* expression, then native membrane extraction by SMA	SMALP	Binding to small red BODIPY-linked ligand	*τ*_D_ = 0.63 ± 0.02 ms *D* = 30 ± 4 µm^2^/s	Grime et al., 2020 [91]
Neurotensin receptor 1 (NTSR1)	*H. sapiens*	HEK293T expression, then native membrane extraction by SMA	SMALP	mRuby fusion construct	τ_D_ = 5.88 ms *D* = 16 µm^2^/s (broad size distribution and large residuals noted)	Dathe et al., 2019 [46]
Wsc1	*S. cerevisiae*	*S. cerevisiae* expression, then native membrane extraction by SMA	SMALP	GFP fusion construct	*D* = 50 ± 4.6 µm^2^/s	Voskoboynikova et al., 2021 [92]
Cytochrome *c* oxidase (Cyt*c*O)	*R. sphaeroides*	Membrane protein from *R. sphaeroides*, MSP from *E. coli*, then mixed with lipids	DOPG + MSP1D1 or MSP1E3D1	Fluorescein on lipids or protein		Xu et al., 2016 [93]

Cell-free co-translation: proteins expressed in the presence of lipids for self-assembly of nanodiscs during expression. SMALP: styrene maleic acid lipid particles. ApoΔ49A1: truncated apolipoprotein A1. DMPC: 1,2-dimyristoyl-sn-glycero-3-phosphocholine. DOPC: dioleoylphosphatidylcholine. DOPE: dioleoylphosphatidylethanolamine. DOPG: dioleoylphosphatidylglycerol. CL: cardiolipin. *τ*_D_: diffusion time. *D*: diffusion coefficient.

## Data Availability

Data presented is contained within this paper.

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
