# Peer review of "Biophysical Characterization of Membrane Proteins Embedded in Nanodiscs Using Fluorescence Correlation Spectroscopy"

_membranes, 2022, doi:10.3390/membranes12040392_

Round 1

Reviewer 1 Report

In this work, Laurence and colleagues summarize the use of FCS spectroscopy to observe and quantify binding events to membrane proteins embedded in nanodiscs. This review is well laid out and provides a clear introduction to FCS for newbies. With a selection of studies, the authors illustrate the biological insights that can be obtained by combining FCS measurements with reconstitution of membranes proteins in nanoliprotein particles.

I only have two comments:

1. The word “nanolipoprotein particle” should be defined clearly be the authors in the context of this review. Because of the variety of systems currently existing to reconstitute membrane proteins, it is easy to be confused. My understanding is that nanolipoprotein particles in this case only regroups polymer-based discs, apoprotein based discs, saposin based discs and variations thereof, and excludes ampiphols, peptidiscs and discs that do not maintain a lipid environment. This should be clarified.

2. FCS is compared with FRAP and FRET but these two methods should be described briefly.

Reviewer 2 Report

In this manuscript, the authors review the isolation of membrane proteins in nanodiscs and their subsequent characterization by fluorescence correlation spectroscopy. Overall, the review is well written, and informative also for non-specialists. Recent examples allow to understand the andvantages of this technique for the study of membrane proteins.

In my opinion, however, a small paragraph on the limitations of the FCS technique for the study of membrane proteins is missing. Problems, such as photobleaching are barely mentioned in the manuscript.

Also, the authors emphasize that nanodiscs are a good alternative to vesicles which contain an indeterminate number of proteins (lines 241-242). I am not sure, however, that the number of embedded proteins can be that  easily controlled in nanodiscs. Several studies referenced by the authors (for instance reference 71 and 80) clearly mention some heterogeneity in the sizes of the particles studied. I would thus suggest to author to add also a small discussion on this important aspect.

Round 2

Reviewer 2 Report

I thank the authors for having taken into consideration all my suggestions. I now believe that the manuscript is suitable for publication in Membranes.